# Effect of Slope Grain on Mechanical Properties of Different Wood Species

**DOI:** 10.3390/ma13071503

**Published:** 2020-03-25

**Authors:** Przemysław Mania, Filip Siuda, Edward Roszyk

**Affiliations:** Department of Wood Science and Thermal Technics, Faculty of Wood Technology, Poznań University of Life Sciences, Wojska Polskiego 38/42, 60-627 Poznań, Poland; filip.siudaa@gmail.com (F.S.); edward.roszyk@up.poznan.pl (E.R.)

**Keywords:** slope of grain, modulus of rupture, work to the maximum load

## Abstract

The aim of the presented study is to determine the relationship between mechanical parameters of selected wood species (*Carya* sp., *Fagus sylvatica* L., *Acer platanoides* L., *Fraxinus excelsior* L., *Ulmus minor* Mill.) used for the production of hand tools and drumsticks and the grain deviation angle from the rectilinear pattern. Modulus of rupture (MOR), modulus of elasticity (MOE), elastic strain and work to maximum load (WML) in the three-point bending test were determined. The results obtained show that the values of all the mechanical parameters measured for hickory wood are higher than those obtained for domestic species. As the grain deviation angle from parallelism increases, the mechanical properties of all analyzed wood species decrease. The greatest influence of grain deviation angle on mechanical parameters was recorded for the work to maximum load values.

## 1. Introduction

The only source of power to manipulate hand tools is the human hand and arm. The primitive man began his experience using stones as tools and over time, learned to give them simple shapes. As their complexity increased, so did their use and the level of technology available to the primitive man. The first stone tools of this type were probably a hammer, an axe, a knife, a hand saw and a drill [1] and they were most likely invented in this order. The handles of the mentioned tools had to be made of a high-quality material having appropriate mechanical parameters. Among the most frequently mentioned species used for the production of everyday items in the Bronze Age in Austria are spruce, beech, ash and birch [2,3].

The modulus of elasticity (MOE) is a basic indicator of high mechanical strength. It can be used for evaluation of the suitability of a given material for the production of tool handles. The best quality hammer and axe grips are made of American hickory (*Carya*) wood. However, they can also be made of species such as European ash (*Fraxinus excelsior*) or European beech (*Fagus sylvatica* L.) and less often, of elm (*Ulmus minor* Mill.) or European hornbeam (*Carpinus betulus* L.).

A hazardous phenomenon resulting from the use of hand tools is the transmission of vibrations due to e.g., impact of these tools. A study on the vibration properties of wood parallel to the grain conducted by Obataya et al. [4] confirms that wood characterized by a large deviation angle of fibrils in the fibers cell wall transmits acoustic waves much worse. The results of Horn’s studies carried out in 1978 [5] confirm that the deviation angle in the wood fibers cell wall of the American hickory (*Carya ovata* species) is much higher than in other deciduous species studied, on average by about 7°.

In addition to vibrations, the human wrist can also be exposed to sudden impacts that directly transfer forces to the wrist muscles and bones. It can happen, for example, when the tool handle breaks and when all impact energy is accumulated in the tool grip and then released in an instant. A common reason for cracking the wooden handles of hand tools is that the wood fibers do not run parallel to their axis.

The slope of grain (SoG) is a deviation of wood fibers from a line parallel to an edge of sawn timber. On the macroscopic scale, variability in wood mechanical properties can be mainly attributed to the slope of grain and wood density. Grain deviation from the directions of the forces causes a decrease in strength [6,7,8]. It has a much higher adverse effect on tensile strength parallel to the grain than on compressive strength or bending strength [9]. A deviation of 15° decreases the tensile strength by more than 50% in comparison with the strength of wood with a straight grain arrangement. A strength reduction due to the increase in the grain deviation angle was also observed in the shearing test. It was reported that a shift in the grain deviation angle from 0 to 30° causes a drop in wood shear strength by about 30%–45% [10,11]. Other authors observed an even more significant decrease in strength, reaching even about 70% [12]. Many authors also indicated the influence of SoG on the modulus of rupture or modulus of elasticity [13,14,15,16]. Given the above, the purpose of this study was to determine the effect of the slope of grain on the mechanical properties of various wood species, particularly as regards elastic energy and work to maximum load. The impact of SoG on these parameters has not yet been described in the context of the suitability of wood, e.g., for the production of drumsticks or tool grips.

## 2. Materials and Methods

Five species of deciduous wood were used in the study: ring-porous, such as hickory (*Carya* sp.), elm (*Ulmus minor* Mill.), ash (*Fraxinus excelsior* L.) and diffuse-porous, such as beech (*Fagus sylvatica* L.) and maple (*Acer platanoides* L.). These species are commonly used for the production of hammer or axe handles, drumsticks or even baseball bats. Elm, ash, beech and maple wood from the Murowana Goslina Experimental Forest Station (geographical coordinates: N 52°32′40.797″; S 17°4′5.132″), west of Poland were used in the study. Planed boards with dimensions of 1000 mm × 250 mm × 30 mm (in the longitudinal (L), radial (R) and tangential (T) direction, respectively) and radial annual ring orientation were cut from butt end heartwood and used for specimen preparation. Hickory wood board was obtained from a drumstick company. Determination of mechanical parameters was carried out in a static bending test on samples with dimensions of 10 (T) × 10 (R) × 150 (L) mm. The use of such samples allowed for more accurate grain orientation and obtaining a grain angle deviation different from parallelism. The samples were cut out in a way that their grain deviation angle was 0, 5, 8, 12 and 15°, respectively (Figure 1). At least ten samples of each species and each variant of grain deviation angle were prepared.

Before measurements, all the samples were conditioned at 20 °C and relative humidity (RH) of 50 ± 2% for two months. Their density was determined according to the method recommended by ISO 13061-2:2014 [17]. The mass of each sample was measured on an analytical balance (Sartorius GmbH, Germany) (±0.001 g accuracy). The dimensions were measured using a digital caliper with an accuracy of ±0.01 mm.

Experimental tests were made using the ZWICK ZO50TH wood testing machine. The integrated software allowed to calculate the modulus of elasticity (MOE), elastic strain energy (U_E_) and work to maximum load (WML), as well as modulus of rupture (MOR) or deformation at the time of destruction. Standards applied for measurements were PN-77/D-04103 [18] for modulus of rupture (MOR) and PN-63/D-04117 [19] for modulus of elasticity (MOE), respectively. The distance between supports during the experiment was 120 mm. The load was applied in the midway of the sample, on the radial surfaces. The rate of loading was chosen in a way to complete the test in about 90 s.

Modulus of rupture (MOR) was calculated as follows:(1)$$MOR=\frac{3F_{max}L}{2bh^{2}}\;(\mathrm{MPa}),$$
where:

*F_max_* – maximum (breaking) force (N),

*L* – the distance between supporting span (mm),

*b*, *h* – width and height of the test samples (mm).

Modulus of elasticity (MOE) was calculated according to the equation:(2)$$\mathrm{MOE}=\frac{3\left(F_{n+1}-F_{n}\right)L^{3}}{64bh^{3}(f_{n+1}-f_{n)}}\;(\mathrm{MPa}),$$
where:

*F_n_*_+1_ − *F_n_* – the increment in load within the linear region of the load-deflection curve (N),

*f_n_*_+1_ − *f_n_* – the increment in deflection (corresponding to *F_n_*_+1_ − *F_n_*), (mm).

Elastic strain energy is defined as the work done in deforming a sample to the stress-strain proportionality limit. Its value is determined based on the equation:(3)$$\;U_{E}=\frac{F_{p}\times f_{p}}{2}$$
where:

*F_p_* – force at the limit of proportionality (N),

*f_p_* – deflection arrow at the limit of proportionality (mm).

The work to maximum load (WML) defines the amount of energy absorbed by the sample until it is destroyed. Its value is equal to the area under the curve σ-ε and is expressed by the equation:(4)$$\mathrm{WML}=\int_{0}^{\varepsilon }\mathrm{MOR}d\varepsilon_{max}\;(J),$$
where:

*ε_max_* – maximum strain (mm).

Following the adopted sample loading scheme, the value of maximum strain in the outermost sample layers was calculated from the Equation:(5)$$\varepsilon_{max}=\frac{6\times h\times f_{max}}{L^{2}}\;(-),$$
where:

*f_max_* – maximal deflection arrow (mm).

After measurement of bending strength, wood moisture content (MC) was also determined by gravimetric method according to the ISO 13061-1 (2014) standard [20].

The experimental data were analyzed using the DellTMStatisticaTM13.1 software with the analysis of variance (ANOVA). Significance of differences between mean values of the measured parameters was determined using Tukey’s HSD test. The comparison tests were performed at a 0.05 significance level. Same superscripts (e.g., a, b or c) within one species denote no significant difference between mean values of the investigated properties.

## 3. Results and Discussion

The results of moisture content and density for all the studied wood species are presented in Table 1.

As it is clear from Table 1, the MC of all the analyzed samples was similar. It is well known that moisture content and temperature significantly affect wood mechanical properties. Almost all wood mechanical parameters increase when MC is reduced below the fiber saturation point (FSP) (approximately 30% MC) [6]. Therefore, it is essential to determine the mechanical parameters of wood at the same moisture content level.

The density of the analyzed species (Table 1) does not differ from those described by Wagenführ [21]. All the results obtained are within the numerical ranges described in the Wood Handbook [22]. The American hickory wood, which is classified as a species of heavy density, was expected to have the highest average density. Its density amounted to approximately 770 kg∙m^−3^, which slightly exceeded the literature value of 690−760 kg∙m^−3^. Elm wood was characterized by the lowest density among all tested species. In the case of common beech wood, the average wood density was 744 kg∙m^−3^, which definitely exceeds the literature data of 680 kg∙m^−3^ [23]. It results from the fact that the samples used in the experiment were cut from the central zone of the tree trunk a few meters above the root neck, which, in the case of beech, is characterized by an increased density [24]. Moreover, due to the broader annual rings in juvenile wood, the density of deciduous wood species is the highest close to the pith and can be higher by almost 30 kg∙m^−3^ than mature wood [6,25]. The average density of American hickory wood was higher than the average density of beech wood by only about 3.3%, while of that of ash wood by about 15%; of elm by over 26%.

Table 2 shows the results of modulus of rupture (MOR) and modulus of elasticity (MOE), for all the five variants of the grain arrangement in the samples. Each result is an average value calculated for measurements of at least 10 samples. The columns also present statistically significant differences within the tested wood species. Same letter designations in the column mean no statistically significant differences between them.

When analyzing the results obtained for samples with parallel grain arrangement, it is clear that the highest values of MOR and MOE were observed for hickory wood characterized by the highest density. The lowest values were recorded for elm, in which interlocked grain is a common defect. Interlocked grain increases wood resistance to splitting, but decreases its bending strength [22,24].

The data presented in Table 2 also show that wood mechanical parameters decrease with an increase in the grain deviation angle. A 15° increase in the grain deviation angle resulted in a 60% and 56% decrease in strength of ash and hickory, respectively. A similar strength reduction was noted in the diffuse-porous wood, where for both species it was about 45%. Considering the decrease in strength, it is clear that elm is the least susceptible to the deviation of wood grain from the rectilinear pattern. The absolute decrease in its strength in comparison to the sample with an almost rectilinear grain pattern was 38%. The results obtained are in line with literature data. Kretschmann et al. [26] reported a decrease in MOR of about 30% as the grain deviation angle increased by 10°. A slightly smaller decrease in bending strength was described by Cown et al. [9], who showed a decrease in wood strength of about 1.2% with a 1° increase in the grain deviation angle.

The reduction of wood strength presented as a graph of the slope of grain against the modulus of rupture (based on average MOR results) is shown in Figure 2.

The observed decrease in strength along with the increase in grain deviation angle is well described by the linear function, which for different wood species is determined by a different coefficient of determination (R^2^), mostly high above 0.9. The lowest determination coefficient was calculated for ash wood, where a sharp drop in strength was observed for angles of 12° and 15°.

The results of statistical analysis indicate that for all wood species, the differences between the values obtained for parallel grain arrangement and the 5° deviation angle are statistically insignificant. Therefore, it can be stated that a slight grain deviation does not significantly reduce the MOR value. A similar relationship can be observed for the highest values of grain deviation angle. There were no significant differences between the average MOR values for 12° and 15° angles.

Data presented in Table 2 clearly show that the MOE measured for all the studied wood species decreases in a similar way. The highest decrease in MOE was observed for ash wood (as high as 65.5%). A slightly smaller decrease was observed in hickory wood (53%). Once again, elm wood proved to be the most resistant to influence of the grain deviation angle on the mechanical parameters, since the MOE reduction was only 42%. It can be explained by the naturally appearing interlocked grain in this species, as described above. In the diffuse-porous species, an increase in the grain deviation angle by 15° caused a decrease in modulus of elasticity by about 47%.

Table 3 presents the results of elastic energy and work to maximum load determination. The ability of wood to accumulate vast amounts of energy is an indicator of low susceptibility to mechanical damage. The wood of American hickory differs significantly from the other studied species in terms of the elastic energy value, which exceeds them approximately twice. An increase in the grain deviation angle in this species led to a 41% decrease in elastic energy. However, it is noteworthy that, despite this, even the lowest values of elastic energy for hickory at 15° grain deviation angle are higher than the values for other species with parallel grain arrangement. It is known that elastic energy depends on the grain deviation angle from the rectilinear pattern. However, in the case of the American hickory in this study, this relationship proves to be ambiguous. Although the modulus of elasticity and the immediate bending strength for this wood decreased significantly, the elastic energy did not show the same tendency, as the value of elastic energy reached its maximum at an angle of 5° (Table 3).

As it is clear from Table 3, the values of elastic energy do not decrease as significantly as in the case of other tested mechanical parameters (Table 2). In the case of elm wood, an increase in the grain deviation angle by 15° resulted in a reduction of elastic energy by up to 44%. The lowest change in elastic energy was recorded in maple wood. The species also had the lowest value of this parameter.

A similar relationship can be observed when analyzing the results of WML. WML determines how much energy a sample of tested wood can absorb until the applied force destroys it. For both tool handles and drumsticks, this parameter will be crucial because their action is hitting a hard working-surface. The highest values were recorded in hickory wood, both for samples with parallel grain arrangement and those with grain deviation. However, this parameter is the most sensitive to change in the orientation of the sample of all analyzed values. The reduction of the work to maximum load for ash samples was over 68% and 64% for beech samples. Slightly smaller decreases were observed in hickory 55%, maple 52% and elm wood 50%. A similar observation can be made for wood intended for special purposes, e.g., used for the production of bows [27]. Based on the results of the research in this study, it can be concluded that beech wood has a much lower WML than hickory wood (*Carya* spp.). The difference in the value of elastic energy between these species was about 54% in favor of hickory wood.

Statistical analysis revealed that, in most cases, an increase in the grain deviation angle in the range of 0–8° causes a similar effect (i.e., differences in average values are statistically insignificant). Only an increase in the grain deviation angle to over 10° causes the differences between the average values making them statistically significant.

## 4. Conclusions

Based on the research carried out and the analysis of the results obtained, the following conclusions were made:The values of physical and mechanical parameters for American Hickory were significantly higher than for the other studied wood species.The grain deviation angle from the rectilinear pattern has a significant impact on the values of mechanical properties and determines the usability of the material intended for the production of tool grips and drumsticks.The most significant influence of the grain deviation angle on mechanical parameters was recorded in the case of work to maximum load values. For all species, they assumed an absolute decrease in value exceeding 50% of the initial value; for the two examined wood species, this decrease was even over 60%.The least influence of the grain deviation angle on the mechanical parameters was observed for elm wood. However, this species was generally characterized by far lower values of mechanical parameters than other species.During the production of tool grips and drumsticks, the deviation of the slope of grain should be avoided, as this may result in their shorter lifetime.

## Figures and Tables

**Figure 1 materials-13-01503-f001:**
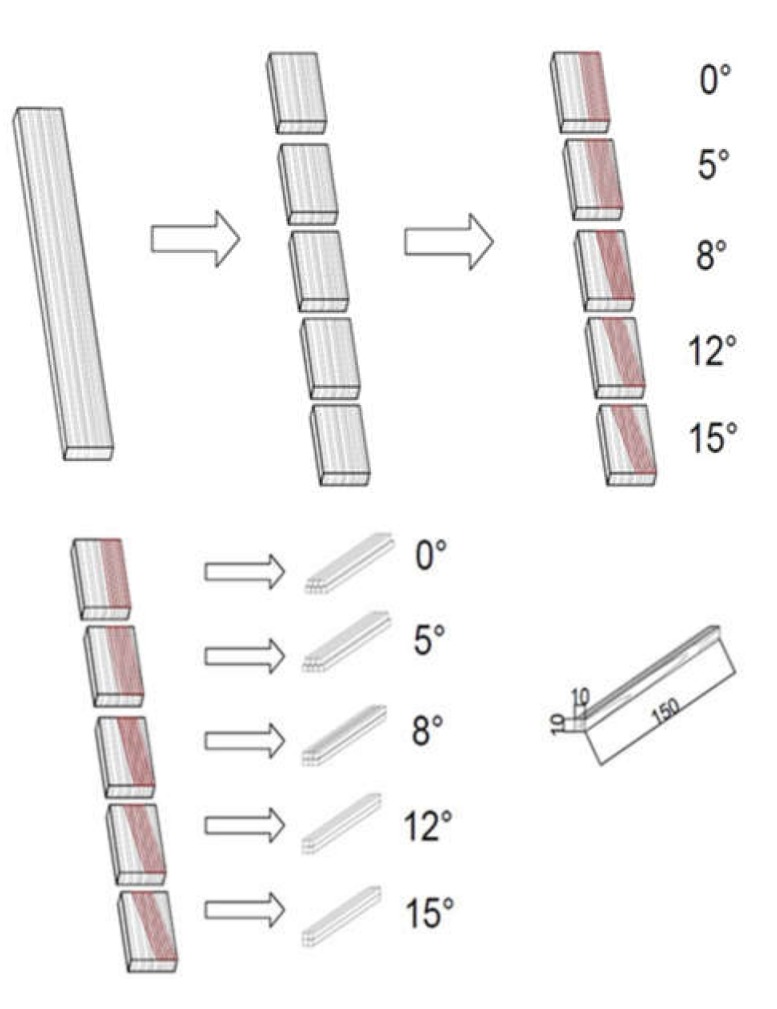
Scheme of cutting the test material into samples.

**Figure 2 materials-13-01503-f002:**
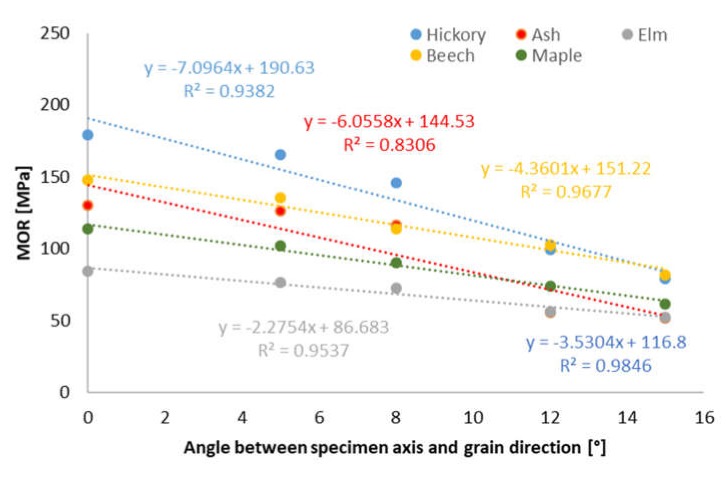
Effect of the slope of grain on the modulus of rupture (MOR).

**Table 1 materials-13-01503-t001:** Moisture content (MC) and wood density of analyzed wood species; ±(SD).

Species	MC (%)	Wood Density (kg·m^−3^)
Hickory	8.9 ± 0.34	770 ± 22
Ash	8.8 ± 0.32	655 ± 16
Elm	8.4 ± 0.41	570 ± 21
Beech	8.9 ± 0.38	745 ± 14
Maple	8.6 ± 0.36	630 ± 17

**Table 2 materials-13-01503-t002:** Mechanical properties of analyzed wood species.

Mechanical Properties	Slope of Grain (°)	Hickory	Ash	Elm	Beech	Maple
Modulus of Rupture MOR (MPa)	0	179.1 ^b^	130.3 ^c^	84.4 ^a^	147.6 ^c^	118.2 ^c^
	5	165.7 ^b^	126.1 ^b,c^	76.9 ^a^	135.5 ^c^	113.7 ^c^
	8	146.1 ^c^	116.6 ^b^	72.5 ^a^	113.7 ^b^	92.6 ^b^
	12	99.4 ^a^	55.6 ^a^	56.1 ^b^	102.9 ^a,b^	82.3 ^a,b^
	15	79.0 ^a^	51.8 ^a^	52.5 ^b^	82.0 ^a^	74.3 ^a^
Modulus of Elasticity MOE (MPa)	0	15380 ^b^	11410 ^c^	7010 ^a^	13440 ^d^	10670 ^b^
	5	14410 ^b^	10790 ^b,c^	6580 ^a^	12030 ^c^	10350 ^b^
	8	12870 ^c^	10390 ^b^	6220 ^a^	12030 ^c^	8360 ^a^
	12	8380 ^a^	4400 ^a^	4810 ^b^	9600 ^a^	7120 ^a^
	15	7260 ^a^	3940 ^a^	4080 ^b^	7000 ^b^	5680 ^c^

^a, b, c, d^ different superscripts denote a statistically significant (*p* < 0.05) difference between mean values according to Tukey’s HSD test.

**Table 3 materials-13-01503-t003:** Mechanical properties of analyzed wood species. Same letter designations in particular columns mean no statistically significant differences between them.

Mechanical Properties	The Slope of Grain (°)	Hickory	Ash	Elm	Beech	Maple
Elastic strain energy U_e_ (J)	0	2.997 ^a^	1.426 ^a^	1.449 ^b^	1.709 ^c^	1.190 ^a^
	5	3.258 ^a^	1.369 ^a^	1.444 ^b^	1.554 ^b,c^	1.179 ^a^
	8	3.028 ^a^	1.297 ^a^	1.191 ^a,b^	1.486 ^a,b,c^	1.157 ^a^
	12	2.060 ^b^	0.862 ^b^	0.864 ^a^	1.149 ^a,b^	1.004 ^a^
	15	1.776 ^b^	0.882 ^b^	0.808 ^a^	1.041 ^a^	0.812 ^b^
Work to maximum Load WML (J)	0	3.574 ^a^	2.722 ^a^	1.505 ^b^	2.687 ^c^	1.730 ^a^
	5	2.957 ^a^	2.715 ^a^	1.203 ^a,b^	2.291 ^b,c^	1.642 ^a^
	8	2.497 ^a^	2.413 ^a^	1.145 ^a,b^	1.922 ^a,b,c^	1.315 ^b^
	12	1.634 ^b^	0.883 ^b^	0.774 ^a^	1.589 ^a,b^	1.209 ^b^
	15	1.614 ^b^	0.837 ^b^	0.746 ^a^	0.963 ^a^	0.836 ^c^

^a, b, c,^ different superscripts denote a statistically significant (*p* < 0.05) difference between mean values according to Tukey’s HSD test.

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
