# Peer review of "Effect of Slope Grain on Mechanical Properties of Different Wood Species"

_materials, 2020, doi:10.3390/ma13071503_

Round 1

Reviewer 1 Report

The manuscript does not provide much new knowledge except the properties of the specific wood species studied which are in fact relevant for tool sticks. Most references are made to very old literature, the present state of the art is not properly described. The static bending test used must be questioned, because an impact bending test would reflect the situation of tool sticks much better. The English needs significant improvement before the paper can be published.

Abstract: include the description of the test set up

L14: instead of native species domestic species may be a better wording

L21: complicated sentence, I suggest to start: The only source of power to manipulate hand tools is the human hand and arm.

L22: replace which by and

L30f: add reference, for example search for younger literature by Grabner and Klein on wood species used in historic tools

L34 delete ‘by Japanese scientists’

L36: replace carries by transmits

L37: stick to citation style and avoid forenames of authors

L39: delete ‘based on’, write complete botanical names in italic

L41: delete ‘prolonged exposure to’, add ‘human wrist’

L46f: These two sentences give two different definitions for slope of grain, use only one if possible with respect to relevant standards

Introduction: what is the gap in knowledge that is addressed by your work?

L59f: separate ring porous from diffuse porous wood species

L63: I don´t under stand the sentence ‘were from the edges coming form the butt end zone’

L64: why did you decide to use the static bending test, when the impact bending test is more relevant for this application?

climate and time for conditioning the samples are missing

L76: replace which by ‘with software that’

L92: replace ‘by’ by ‘to deform’

L102: isn´t sigma max the MOR from Equ. 1?, add a definition/formula for epsilon max

L104: change word order to make a readable sentence (After….MC was  determined by ….), add the measurement on control samples

L110: add ‘of’ the investigated properties

Table 1: add standard deviations

L116 move method description to the right chapter

L118: replace ‘both as’ by ‘when’, add references for this statement

L120f: delete statement on 50% reduction because this is not a result of this study

L122: replace ‘bent’ by ‘tested’

L123: do you mean MC instead of humidity?

L130: add ‘of’ 680

L133: add references on density in juvenile wood of hardwood species

L136f: change word order to make a readable sentence

Table 2: correct ‘analysed’, add a clear description of significance indicators

L145: use defined parameters MOR and MOE

L146: replace characterized by obtained

L153: avoid ‘you can see’

Fig. 2: describe linear regression analysis in methods chapter, was this done with mean values or with single test results?

L166: was the regression analysis done with mean values or with single test results?

L169ff: can the result of linear regression be significant when the differences of means between the low and high grain deviation groups were not significant?

L174f: use ‘also’ only once

L176: delete ‘it’

Table 3: correct ‘analysed’, add a clear description of significance indicators

L205: avoid ‘respectively’ and write values next to the names

L207: use the defined WML instead of ‘energy of destruction’

L214ff: delete comments from earlier reviews

Reviewer 2 Report

The manuscript provide new knowledge and interesting information for the research/industry community. Nevertheless, there are several changes I would like to recommend for the improvement of the quality:

In introduction you should provide more background information to the reader on the topic, presenting more previously published scientific papers. Generally, in my opinion the introduction chapter is not very well prepared/written and you should make some changes/additions in order to provide an improved quality of writing. For example, you presented the aim and objectives of this work in only one line, which of course does not highlight the incentive and significance of the work.

In line 44, pleas explain the phrase " they are not made properly".

In line 54, check again the word "angel", and a detailed check for grammatical and syntactical errors in the whole text should be implemented, since several grammatical/syntactical errors were detected (for example lines 112, 113, 128, 130, 142, 144, 146 etc.).

In Materials chapter, you do not refer the reason why you chose the specific species in the experiment. It would be necessary to include significant information such as how many trees where cut/used for each species, their origin, which was the sampling method you based your experiment on, meaning how you chose to cut the initial boards from the stem, which heights, how many different heights etc., age of trees, diameters, wood densities, 

Figure 1 is very descriptive and useful. It is necessary also to provide an image of the final specimens, showing the result appearance of the different slopes. 

For the test of modulus of elasticity (MOE), elastic strain energy (UE) and
work to maximum load (WML), modulus of rupture (MOR) or deformation at the time of destruction, you should provide the standards used.

In results, in table 1, you do not present the standard deviation of the values. Could you explain how is it possible, under the same conditions of conditioning, all the species to present so similar value of MC, when the densities are so different one another. Please, check again your values. Do you refer to MC or you mean probably equilibrium moisture content EMC? 

118-119 lines and 120-121 lines: provide the reference. Did you conduct tests also in above saturation point?

You rather change the phrase "mechanical parameters" to mechanical properties or strength, since the meaning sometimes changes.

123 line: not humidity, correct to moisture content. To which moisture content do you refer at this point. It is not clear.

In lines 132-134: Interpreting the results of high density for beech wood, you use the word core. Do you want to refer to the pith, the heartwood part or the central part regarding the height? Please, check again the terminology used.

134 line: provide reference.

193 and 210 line: there is not table 4.

Table 2 and table 3 should include also standard deviation values and not only the mean values. Around the table you should explain again the letters a,b,c to be useful to the reader. 

Line 199: you referred this earlier in the text.

214-216 lines: Please, erase these comments. 

The conclusions is a very good summary of the results but do not provide something more than that. You can add some phrases to highlight the significance/soundness of the work. 

Round 2

Reviewer 1 Report

The manuscript improved by the revision of the autors. A few remarks remain for minor revision before the paper can be published.

L46f: These two sentences still  give two different definitions for slope of grain: the first sentence is a definition and within the brackets of the second sentence there is another definition, I suggest to delete the part in brackets

L111: then replace sigma by MOR in Equ. 1 and the legend

Table 2: add a clear description of significance indicators to the table captions

L170: the regression analysis should be done with single test results to analyse the real distribution of data.

L173ff: can the result of linear regression be significant when the differences of means between 0° and 5° as well as between 12° and 15° were not significant?

References 5, 6, 8, 13, 14, 15, 17, 18, 19, 20, 22, 23, 24: years should be printed in bold; the new references 13 to 15 are again very old literature as many of the other citations

Reference 15: year must be corrected
